# Enhanced Hepatogenic Differentiation of Human Wharton’s Jelly–Derived Mesenchymal Stem Cells by Using Three-Step Protocol

**DOI:** 10.3390/ijms20123016

**Published:** 2019-06-20

**Authors:** Wachira Panta, Sumeth Imsoonthornruksa, Ton Yoisungnern, Sanong Suksaweang, Mariena Ketudat-Cairns, Rangsun Parnpai

**Affiliations:** 1Embryo Technology and Stem Cell Research Center, School of Biotechnology, Institute of Agricultural Technology, Suranaree University of Technology, Nakhon Ratchasima 30000, Thailand; wachira@medezegroup.com (W.P.); tonikuya@hotmail.com (S.I.); tonton9214@gmail.com (T.Y.); ketudatcairns@hotmail.com (M.K.-C.); 2School of Pathology and Laboratory Medicine, Institute of Medicine, Suranaree University of Technology, Nakhon Ratchasima 30000, Thailand; ssnnkk@yahoo.com

**Keywords:** endoderm, Wharton’s jelly mesenchymal stem cells, hepatocyte-like cells, histone deacetylase inhibitor, sodium butyrate

## Abstract

Currently, human Wharton’s jelly-derived mesenchymal stem cells (hWJ-MSCs) are an attractive source of stem cells for cell-based therapy, owing to their ability to undergo self-renewal and differentiate into all mesodermal, some neuroectodermal, and endodermal progenies, including hepatocytes. Herein, this study aimed to investigate the effects of sodium butyrate (NaBu), an epigenetic regulator that directly inhibits histone deacetylase, on hepatic endodermal lineage differentiation of hWJ-MSCs. NaBu, at 1 mM, optimally promoted endodermal differentiation of hWJ-MSCs, along with epidermal growth factor (EGF) and basic fibroblast growth factor (bFGF) supplementation. *CXCR4*, *HNF3β*, *SOX17* (endodermal), and *GATA6* (mesendodermal) mRNAs were also up-regulated (*p* < 0.001). Immunocytochemistry and a Western blot analysis of SOX17 and HNF3β confirmed that the 1 mM NaBu along with EGF and bFGF supplementation condition was appropriately pre-treated with hWJ-MSCs before hepatogenic differentiation. Furthermore, the hepatic differentiation medium with NaBu pre-treatment up-regulated hepatoblast (AFP and HNF3β) and hepatic (CK18 and ALB) markers, and increased the proportion of mature hepatocyte functions, including *G6P*, *C/EBPα*, and *CYP2B6* mRNAs, glycogen storage and urea secretion. The hepatic differentiation medium with NaBu in the pre-treatment step can induce hWJ-MSC differentiation toward endodermal, hepatoblastic, and hepatic lineages. Therefore, the hepatic differentiation medium with NaBu pre-treatment for differentiating hWJ-MSCs could represent an alternative protocol for cell-based therapy and drug screening in clinical applications.

## 1. Introduction

The liver is the largest organ and essentially produces metabolites and detoxifying toxic substances. Hepatocytes can be gradually destroyed by toxic substances, which can lead to liver failure and death at the end-stages of liver diseases [1]. For such patients, treatments including orthotropic organ and cell transplantations have recently been pioneered. Nevertheless, the shortage of organ donors and low quality of cultivated hepatocytes remain the major problems associated with these types of treatments [2,3]. Owing to these limitations, new alternative therapeutic approaches are required. Stem cell technology is considered a promising tool for cell-based therapy. Thus, mesenchymal stem cell (MSC) biology has recently received increasing attention.

Compared to pluripotent embryonic stem cells (ESCs), MSCs are associated with minimal ethical concerns and result in fewer teratomas, thus, they represent a promising tool for regenerative medicine. In addition, MSCs have certain attractive qualities such as plastic attachments, self-renewal, positive expression of unrestricted MSC markers, and tri-mesodermal differentiation to lineages such as osteoblasts, chondrocytes, and adipocytes [4]. MSCs can be isolated from fetal, adult, and postnatal tissues such as bone marrow, adipose tissue, amniotic fluid, umbilical cord blood, and umbilical cord tissue [5,6,7]. Hence, MSCs are promising for therapeutic and clinical applications.

Umbilical cord Wharton’s jelly tissue is the richest and most attractive source of MSCs, as it can be obtained through non-invasive methods. It also has lower immunogenicity than other adult MSC tissue sources [8]. These cells can differentiate into hepatocyte-like cells of the endodermal lineage, making them an alternative for studies on hepatocyte-like cell differentiation [9,10]. Many in vitro hepatogenic differentiation protocols for hWJ-MSCs have been successfully implemented [11,12,13,14,15,16]; however, growth factors or cytokines or chemicals with greater efficiency are needed to obtain homogenous hepatocyte-like cell populations.

Sodium butyrate (NaBu) is a small molecule, short-chain sodium salt of butyric acid [17]. It essentially functions as a histone deacetylase inhibitor (HDACi), and can be used to inhibit tumorigenesis [18], treat neurological disorders [19], and induce stem cell differentiation at physiological concentrations [20]. It has been used successfully and consistently to induce the differentiation of ESCs to hepatocyte-like cells [21,22,23,24,25]. Therefore, NaBu is considered an appropriate choice to promote similar MSC differentiation.

The aim of this study was to confirm this approach for the differentiation of hWJ-MSCs into functional hepatocyte-like cells that express endodermal, hepatoblastic, and hepatic markers. To achieve this objective, the effect of NaBu in the pre-treatment step (for 3 days) with and without epidermal growth factor (EGF) and basic fibroblast growth factor (bFGF) was investigated. This was followed by the induction of hepatogenic differentiation in hWJ-MSCs, using other maturation factors [11] such as hepatocyte growth factor (HGF), bFGF, and nicotinamide (for the differentiation step) and oncostatin M (OSM), dexamethasone, and insulin-transferrin-selenium-ethanolamine (ITS) (for the maturation step). This protocol can provide an alternative source of hWJ-MSC-derived hepatocyte-like cells for stem cell-based therapy in patients with end-stage liver diseases.

## 2. Results

### 2.1. Isolation and Characterization of hWJ-MSCs

hWJ-MSCs were isolated after performing as previously described [26] by using a tissue explant procedure. Cells that migrated from the tissue displayed fibroblast-like morphology in culture (Figure 1A) and approached 80% confluence in 7 days (Figure 1B). These cells were harvested and grown for further use.

hWJ-MSCs (*n* = 3) were assessed at passages 3–7 via growth kinetics of the cell number, cumulative population doubling level (CPDL), and population doubling time (PDt) (Appendix A). Moreover, hWJ-MSCs were characterized at passage 4 via immunophenotyping and multipotency assays (Appendix A). Hepatogenic differentiation of hWJ-MSCs were induced by using the modified standard protocol with the previous study [11] and characterized by using immunofluorescence (alpha-fetoprotein; AFP and albumin; ALB) and Periodic acid-Schiff (PAS) staining (Appendix A). Among hWJ-MSCs #1, hWJ-MSCs #2, and hWJ-MSCs #3, it was found that immunophenotyping and multipotency properties did not perform different patterns, while growth kinetics and hepatic differentiation by using immunofluorescence and PAS staining of hWJ-MSCs #3 were greater exhibited expressions of hepatic-specific features than hWJ-MSCs #1 and hWJ-MSCs #2. Therefore, the authors choose hWJ-MSCs #3 in hepatogenic differentiation for further study by using a three-step protocol of induction.

An immunocytochemical analysis revealed that hWJ-MSCs positively expressed the MSC markers CD73, CD90, and CD105, whereas CD34, a hematopoietic marker, was not detected (Figure 1C (a–d)). The in vitro tri-mesodermal lineage differentiation potential of hWJ-MSCs were examined at day 21 after induction by Alizarin Red, Alcian Blue, and Oil Red O staining for osteogenic, chondrogenic, and adipogenic lineages, respectively. Differentiated cells exhibited calcium mineralization (Figure 1D (a)), proteoglycan matrix production (Figure 1D (b)), and intracytoplasmic lipid droplet formation (Figure 1D (c)), characteristic of osteoblast, chondroblast, and adipocyte lineages, respectively. These data indicate that the hWJ-MSCs have typical MSC characteristics.

### 2.2. Effect of NaBu Treatment on hWJ-MSC Viability

To examine the cytoxicity of NaBu, hWJ-MSCs were cultured in serum-free medium supplemented with NaBu at various concentrations (0, 1, 2.5, 5, and 10 mM) for 72 h and cell survival was quantified via MTT assays. Supplementation with 1 mM NaBu resulted in significantly higher cell viability (98.39 ± 4.85%) when compared to 2.5, 5, and 10 mM NaBu (81.77 ± 6.94%, 79.01 ± 5.46%, and 53.37 ± 6.34%, respectively) (** p* < 0.05) (Figure 2). These data indicate that 1 mM NaBu can be used for hepatogenic differentiation during pre-treatment.

### 2.3. Effect of NaBu on Epigenetic Statuses and Endodermal Differentiation of hWJ-MSCs

The study next investigated the morphological changes and endodermal gene and protein expression of hWJ-MSCs after treatment with NaBu at various concentrations (1–5 mM) with and without EGF and bFGF supplementation for 3 days. hWJ-MSCs after only NaBu (1–5 mM) treatment became flatter than the control (Figure 3A (a–e)). However, the (1–5 mM) NaBu with and without EGF and bFGF conditions yielded spindle-shaped cells similar to the control (Figure 3A (f–i)). Compared with control hWJ-MSCs, treatment of hWJ-MSCs with the 1 mM NaBu along with EGF and bFGF condition enhanced the significantly highest expression of definitive endodermal specific genes such as *CXCR4* (88 fold), *SOX17* (33 fold) and *HNF-3β* (9 fold) (^Δ^
*p* < 0.001, *** *p* < 0.001 vs. ^##^
*p* < 0.01 and ^###^
*p* < 0.001) on RT-PCR. Additionally, hWJ-MSCs exposed to the 1 mM NaBu along with EGF and bFGF condition also enhanced the significantly highest expression of mesendoderm *GATA6* mRNA expression (1.5 fold) (Figure 3B) when compared to other conditions, except the control group (^Δ^
*p* < 0.001, * *p* < 0.05 vs. ^##^
*p* < 0.01 and ^###^
*p* < 0.001).

To further analyze endodermal differentiation, the authors performed immunofluorescence staining and western blot analyses for definitive endodermal markers, SOX17 and HNF-3β. SOX17 and HNF3β were up-regulated in the 1 mM NaBu along with EGF and bFGF condition-treated cells compared to the control (Figure 4A (a–l)). Among all treatments, the percentage of cells expressing SOX17 and HNF3β in the 1 mM NaBu along with EGF and bFGF condition were distinctly higher than those in other conditions (39.60 ± 0.08% and 52.47 ± 0.09%, respectively) (^Δ^
*p* < 0.001, * *p* < 0.05, ** *p* < 0.01 and *** *p* < 0.001 vs. ^#^
*p* < 0.05 and ^###^
*p* < 0.001) (Figure 4B). Additionally, SOX17 and HNF3β were confirmed via a western blot analysis. As shown in Figure 4D, both SOX17 and HNF3β protein expression were greater in cells cultured in control, EGF and bFGF, and 1 mM NaBu along with EGF and bFGF conditions. Therefore, hWJ-MSCs cultured in 1 mM NaBu along with EGF and bFGF condition committed to endodermal lineage differentiation, appropriate for further hepatogenic differentiation.

The effect of NaBu pre-treatment through HDAC1 inhibition and the H4K5^Ace^ statuses were investigated during endodermal differentiation of hWJ-MSCs in a dose-dependent manner. hWJ-MSCs were pre-treated with NaBu at various concentrations (0–5 mM) and supplemented with EGF and bFGF. Accordingly, the 1 mM NaBu along with EGF and bFGF condition showed decreased HDAC1 protein expression, as shown by immunofluorescence (Figure 4A (m-r)) and western blot analyses (Figure 4D). However, mRNA levels of *HDAC1* in the 1 mM NaBu along with EGF and bFGF condition were somewhat lower; however, statistical significance was not achieved when comparing to that in the EGF and bFGF conditions (^Δ^
*p* < 0.01, * *p* < 0.05 vs. ^##^
*p* < 0.01) (Figure 4C). Moreover, H4K5^Ace^ was more strongly detected at various concentrations of NaBu (0–5 mM) pre-treatment with and without EGF and bFGF supplementation, including the control (Figure 4D). These results indicate that hWJ-MSCs could differentiate into the endodermal lineage through histone acetylation of epigenetic modifications when cultured in the presence of 1 mM NaBu along with EGF and bFGF condition. This result also confirmed that the 1 mM NaBu along with EGF and bFGF condition can be used for hepatogenic differentiation of these cells.

### 2.4. Effect of NaBu on Hepatogenic Differentiation of hWJ-MSCs

This study further assessed hWJ-MSC differentiation into hepatocyte-like cells, during 3 days pre-treatment with the optimized concentration of 1 mM NaBu combined with EGF and bFGF, before performing the further 2-step hepatic lineage differentiation protocol of Campard and colleagues with some modifications [11]. The morphological changes were assessed on days 3, 10, 17, and 24 following initiation of the protocol. Upon differentiation, the authors observed morphological changes from fibroblast-like cells to spherical or polygonal cells with hepatocyte features (Figure 5). At day 3, the fibroblast-as morphology of hWJ-MSCs had not changed in both groups (the hepatic differentiation medium with and without NaBu pre-treatment) when compared to the control (Figure 5A (a–c)). On day 10 of differentiation, cells acquired a more spindle-like morphology (Figure 5A (d–f)), which continuously changed into an epithelial-like spherical shape by day 17 (Figure 5A (g–i)). The cells became polygonal on day 24 (Figure 5A (j–l)). These morphological changes occurred in both groups of differentiated hWJ-MSCs (the hepatic differentiation medium with and without NaBu pre-treatment).

To analyze hepatogenic differentiation of hWJ-MSCs, a RT-PCR analysis was performed for hepatic markers such as *AFP*, *HNF3β*, *CK18*, *ALB*, *G6P*, *C/EBPα*, and *CYP2B6* on days 3, 10, 17, and 24 following the initiation of differentiation (Figure 5B). During the early stages (within 10 day) of differentiation, significantly higher *AFP* and *HNF3β* transcript levels (29-fold and 13-fold, respectively) were detected in the hepatic differentiation medium with NaBu pre-treatment condition than in the group of hepatic differentiation medium without NaBu pre-treatment (16-fold and 6-fold, respectively) or the control group (^Δ^
*p* < 0.01 vs. * *p* < 0.05, ** *p* < 0.01, and *** *p* < 0.001). *AFP* genes were gradually down-regulated in both the hepatic differentiation medium with and without NaBu pre-treatment conditions on days 17 (12-fold and 7-fold, respectively) and 24 (4-fold and 0.5-fold, respectively) (* *p* < 0.05, ** *p* < 0.01, and *** *p* < 0.001) of differentiation. However, the expression of *HNF3β* mRNA in the hepatic differentiation medium without NaBu pre-treatment was highest on day 17 (10-fold) and gradually decreased on day 24 (3-fold) (*** *p* < 0.001). Regarding the mRNA expression patterns of *CK18* and *ALB* (Figure 5B), RT-PCR analysis suggested that hWJ-MSCs could be differentiated into hepatocyte-like cells. During the early stage (within 10 days) of differentiation, the mRNA expressions of *CK18*, *ALB*, *G6P*, *C/EBPα*, and *CYP2B6* were higher in the hepatic differentiation medium with NaBu pre-treatment condition (3-fold, 4-fold, 3-fold, 7-fold, and, 2-fold, respectively) than in the hepatic differentiation medium without NaBu pre-treatment (2-fold, 3-fold, 2-fold, 4-fold, and 1-fold, respectively) and control groups (* *p* < 0.05, ** *p* < 0.01, and *** *p* < 0.001). On day 17 of differentiation, differentiated hWJ-MSCs in the hepatic differentiation medium with NaBu pre-treatment condition showed increased *CK18*, *ALB*, *G6P*, *C/EBPα*, and *CYP2B6* mRNA expression (* *p* < 0.05, ** *p* < 0.01 and *** *p* < 0.001), and the highest expression levels were detected on day 24 of differentiation (10-fold, 7-fold, 9-fold, 8-fold, and 11-fold, respectively). The levels were also significantly different when compared to the expression of these markers in differentiated hWJ-MSCs in the hepatic differentiation medium without NaBu pre-treatment (3-fold, 4-fold, 6-fold, 5-fold, and 6-fold, respectively) and control groups (^Δ^
*p* < 0.01 vs. * *p* < 0.05 and *** *p* < 0.001).

Interestingly, the authors observed that hWJ-MSCs constitutively express some hepatic markers. Indeed, AFP, HNF3β, CK18, and ALB were detected at the protein level in hWJ-MSCs cultured in the expansion medium (Figure 6A (a,e,i,m)). The percentage of cells expressing hepatic markers of hWJ-MSCs among hWJ-MSCs were 45.92 ± 6.97% (AFP), 2.17 ± 1.95% (HNF3β), 4.68 ± 1.39% (CK18), and 13.67 ± 5.97% (ALB) (Figure 6B). Immunocytochemistry of differentiated hWJ-MSCs in the hepatic differentiation medium with and without NaBu pre-treatment conditions performed that all hepatic markers (AFP, HNF3β, CK18 and ALB) could also detect all during periods of differentiation (Appendix A). Nevertheless, the strongest expression of hepatic markers were detected on day 24 of differentiation (Figure 6A (c–d, g–h, k–l, o–p)) in both the hepatic differentiation medium with and without NaBu pretreatment conditions when compared to the control (Figure 6A (b,f,j,n)). To confirm the previous results, the percentage of cells expressing hepatic markers was assessed in the hepatic differentiation medium with NaBu pre-treatment population and was higher on day 24 of differentiation (90.92 ± 6.30% (AFP), 68.33 ± 7.64% (HNF3β), 90.61 ± 8.40% (CK18) and 96.94 ± 0.69% (ALB)) when compared to the group of the hepatic differentiation medium without NaBu pre-treatment condition ((78.90 ± 10.87% (AFP), 50.46 ± 8.49% (HNF3β), 74.67 ± 7.84% (CK18) and 86.52 ± 7.00% (ALB)) (^Δ^
*p* < 0.001, * *p* < 0.05, ** *p* < 0.01 and *** *p* < 0.001 vs. ^#^
*p* < 0.05, ^##^
*p* < 0.01 and ^###^
*p* < 0.001) (Figure 6B). Additionally, hepatic markers were also confirmed via a Western blot analysis. As shown in Figure 6C, AFP and HNF3β expressions were observed in all stages of differentiation, including hWJ-MSCs. However, protein expressions of CK18 and ALB were detected at mid-late and late stages of differentiation, respectively.

The biochemical functions of mature hepatocytes were assessed in the cytoplasmic glycogen storage granules via Periodic acid-Schiff (PAS) staining, and urea production at days 3, 10, 17, and 24 of differentiation. Undifferentiated hWJ-MSCs were not positive for PAS staining (Figure 7A (a)), however, differentiated hWJ-MSCs in the hepatic differentiation medium with NaBu pre-treatment condition (Figure 7A (c)) showed stronger PAS staining than that in cells cultured in the hepatogenic medium - NaBu pre-treatment condition (Figure 7A (b)). These results correspond to the expression of hepatic markers on days 3, 10, and 17. In particular, increased PAS staining was observed on days 3 and 10, and a stronger signal was detected on day 17 in both groups of differentiated hWJ-MSCs (the hepatic differentiation medium with and without NaBu pre-treatment) (Appendix A). Urea secretion, based on the metabolic function of ammonia detoxification, was investigated on days 3, 10, 17, and 24 (Figure 7B). In both groups of differentiated hWJ-MSCs (the hepatic differentiation medium with and without NaBu pre-treatment), only a marginal increase in urea production was detected on days 3 and 10, when compared to the control group, and this difference was not statistically significant. On days 17 and 24 of differentiation, both groups of differentiated hWJ-MSCs (the hepatic differentiation medium with and without NaBu pre-treatment) displayed increased urea production when compared to the control. Additionally, significant differences were also observed between these groups (^Δ^
*p* < 0.05, * *p* < 0.05, vs. ^#^
*p* < 0.05). Therefore, hWJ-MSCs cultured in the hepatic differentiation medium with NaBu pre-treatement condition can be differentiated into mature hepatocyte-like cells. In addition, these cells were functionally enhanced when compared to those differentiated in the hepatic differentiation medium without NaBu pre-treatment condition.

## 3. Discussion

Currently, transplantation of undifferentiated hWJ-MSCs has been performed to treat degenerative diseases, including those of the liver. This can improve liver function and symptoms of end-stage liver diseases [27]. Based on their properties, hWJ-MSCs are potentially applicable for treating liver diseases. Thus, hWJ-MSC-derived hepatocyte-like cells may be an alternative promising resource for end-stage liver diseases.

In this study, the initial population of MSCs was obtained from human umbilical cord WJ tissue-by-tissue explants in accordance with a previously reported method [26], after removing umbilical blood vessels. Our data indicate that hWJ-MSCs expressed MSC-specific markers including CD73, CD90, and CD105, but not CD34, a hematopoietic cell marker. They were also shown to have tri-mesodermal lineage differentiation potential and could be differentiated into osteoblasts, chondrocytes, and adipocytes in vitro. Therefore, the hWJ-MSCs used in this study have characteristics of hMSC in accordance with the criteria of a previous study [11,26].

Currently, HDACi tricostatin A, valproic acid (VPA), and NaBu have received increasing attention for their ability to suppress cell proliferation, promote cell differentiation, and apoptosis in tumor and stem cells [28,29]. NaBu has been shown to inhibit the proliferation of MSCs [30,31]. In accordance with previous studies and in the present study, 1 mM NaBu only slightly suppressed proliferation of MSCs from umbilical cord blood, WJ, and adipose tissues, and this was not significantly different from the controls [30,32]. Additionally, our data, based on other concentrations (2.5, 5, and 10 mM), revealed significant suppression of cell growth when compared to the control and 1 mM NaBu treatment groups.

To induce MSC differentiation successfully toward hepatocyte-like cells, bFGF, HGF, and nicotinamide play a key role in the early steps of hepatic induction. Furthermore, during the differentiation and maturation stages, OSM, dexamethasone, and ITS have been used for the hepatogenic differentiation of MSCs [11,13,33,34]. Hepatogenesis is influenced by bFGF and HGF during early stages of embryonic endoderm development, while OSM (a member of the interleukin-6 cytokine family) is involved in the primary maturation fate of fetal hepatogenesis [35,36,37,38]. Moreover, nicotinamide is necessary for the proliferation and differentiation of small hepatocyte colonies from rat primary hepatocytes in vitro [39]. Dexamethasone and ITS are required for maintaining the expression of liver-enriched transcription factors (LETFs) and promoting cell survival, respectively, which are essential for stimulating liver-specific gene transcription [15,40]. Based on the primitive origin of hWJ-MSCs, they are more immature than adult MSCs from other sources. Therefore, in previous studies, an increase in the concentration of exogenous HGF (40–50 ng/mL) [12,13] or OSM (40 ng/mL) [15] was considered to trigger the hepatogenic signal transduction.

Increasing evidence indicates that hWJ-MSCs could be differentiated into endodermal-like cells in vitro [16,41]. Recently, NaBu was shown to promote hepatic differentiation of ESCs [21,22,23,24,25]. Nevertheless, no evidence is available regarding the effect of NaBu on hepatogenic differentiation of hMSCs. Hepatogenic differentiation of mouse and rat multipotent adult progenitor cell (MAPC), MSC-like cells, indicates that NaB cannot promote hepatogenic differentiation [42]. However, herein, the authors showed an improved method for the hepatogenic differentiation of hWJ-MSCs by NaBu pre-treatment for 3 days by referring to previous studies on BM-MSC induction [28,29]. Therefore, in this study, the effect of NaBu on hepatogenic differentiation of hWJ-MSCs by NaBu pre-treatment for 3 days with some modifications in the previously established 2-step differentiation protocol [11] was investigated. The expression of mesendodermal and endodermal markers of differentiated hWJ-MSCs were verified using immunofluorescence staining, RT-PCR, and western blot analyses. The 1 mM NaBu along with EGF and bFGF condition was synergistically endodermal (SOX17 and HNF3β) differentiation of hWJ-MSCs at both gene and protein levels, including mesendodermal *GATA6* mRNA level. SOX17 is a high-mobility-group box domain transcription factor, and is essential for definitive endoderm formation [43]. GATA6, a zinc finger transcription factor that plays an important role during embryonic liver development that important for mesendodermal specification, and involved in heart (mesoderm) and liver (endoderm) formation during embryogenesis [44]. HNF3β or FOXA2, essential for liver development initiation, possess DNA-binding domains containing a forkhead box (FOX) helix-loop-helix that includes extended loop or winged helix structures [45]. However, in some studies, they considered definitive endodermal markers of hWJ-MSC differentiation [16,41]. Typically, CXCR4 is expressed in mesodermal progenitors and hematopoietic stem cells [46]. However, previous studies also considered this a marker of endodermal identification of ESC and MSC differentiation [16,47]. Corresponding to mesendodermal and endodermal differentiation of hWJ-MSCs, the 1 mM NaBu along with EGF and bFGF condition promoted the highest expression of *CXCR4* mRNAs than other conditions. In addition, during endodermal differentiation of hWJ-MSCs, the effect of NaBu on epigenetic statuses of HDAC1 activity inhibition and H4K5^Ace^ in parallel was investigated. Our data indicated that HDAC1 activity was inhibited in the 1 mM NaBu along with EGF and bFGF condition, in terms of gene and protein expression. Nevertheless, histone acetylation at histone H4 was not only H4K5^Ace^ in culturing hWJ-MSCs in 1 mM NaBu along with EGF and bFGF condition, but also detected in the EGF and bFGF and with higher concentrations of NaBu pre-treatment (2.5 and 5 mM) conditions, including hWJ-MSCs. These results, corresponding to those of a previous study, suggest that protein expression of histone H4^Ace^ in hepatogenic differentiation of BM-MSCs was detected in both with and without VPA pre-treated conditions on a Western blot analysis [28,29]. Therefore, the authors proposed that the low concentrations of NaBu (1 mM), combined with EGF and bFGF during pre-treatment, could efficiently promote endodermal differentiation of hWJ-MSCs compared to the higher concentrations of NaBu (2.5 and 5 mM), owing to HDAC1 activity being hindered and endodermal lineage specification of hWJ-MSCs not being promoted. Therefore, the epigenetic modifications via HDAC1 inhibition and H4K5^Ace^ of NaBu pre-treatment could synergize with EGF and bFGF to promote hWJ-MSC differentiation toward the endodermal lineage. The 1 mM NaBu along with EGF and bFGF condition was the appropriately chosen pre-treatment of hWJ-MSCs before hepatogenic lineage differentiation. However, the mechanism through which NaBu and EGF/bFGF enhance hWJ-MSC differentiation to endodermal lineages via HDAC1 inhibition and H4K5^Ace^ should be further investigated.

AFP is an early hepatic differentiation marker of immature fetal hepatocytes [12]. At the protein level, our data indicate that the expression of AFP and HNF3β were detected in all stages of differentiation including undifferentiated hWJ-MSCs at various expression levels, via immunocytochemistry and Western blot analyses. Similar to a previous study, Western blot results of Zhang and colleagues indicated that AFP was detected in all stages of hepatogenic differentiation of hWJ-MSCs. However, higher levels were detected in early and intermediate stages of differentiation and gradually decreased in later stages of differentiation [12]. Moreover, immunocytochemistry and the percentage of cells that most strongly expressed AFP and HNF3β were on day 24 when compared with other times. However, an analysis of mRNA expression of *HNF3β* and *AFP* indicated that differentiated hWJ-MSCs in the hepatic differentiation medium with NaBu pre-treatment condition were significantly highest in the hepatic differentiation medium without NaBu pre-treament condition at an early stage (within 10 days) of development. These results correspond to those of previous studies on hepatogenic differentiation of BM-MSCs via VPA pre-treatment [28,29]. Nevertheless, hepatogenic differentiation of hWJ-MSCs in a normal culture system (without HDACi pre-treatment) revealed that the mRNA levels of *AFP* were detected later than in the HDACi pre-treatment culture system, through detection of significantly higher expression in the mid-late stage of differentiation [12,15]. These data indicated that differentiated hWJ-MSCs in the hepatic differentiation medium with NaBu pre-treatment condition indicated enhanced differentiation toward hepatoblast-like cells, compared to those observed in differentiated hWJ-MSCs the hepatic differentiation medium without NaBu pre-treatment condition.

The middle to late stages of liver development, differentiated hWJ-MSCs in the hepatic differentiation medium with and without NaBu pre-treatment conditions demonstrated characteristic gene and protein expression of CK18 and ALB via RT-PCR, immunofluorescence and Western blot analyses. CK18 and ALB have been reported to be mature hepatocyte markers of mid-late and late stages of differentiation during liver organogenesis [12]. This study indicates that *CK18* and *ALB* mRNA expression was significantly higher in the middle and late stages of differentiation. These data are similar to those of a previous study on hepatogenic differentiation of hBM-MSCs by using VPA pre-treatment, the *ALB* mRNA expression level was also significantly higher at the late stage of differentiation [29]. However, in the normal culture system, the mRNA expression of *ALB* was different by showing significantly highest expression at the middle stage and gradually in late stage of differentiation, but the expression of the *CK18* mRNA expression was similar [12]. At the protein level, our data indicate that the expression of CK18 and ALB was detected at mid-late and late stages of differentiation, respectively, in this system via immunocytochemistry, analysis of the percentage of positive cells, and Western blot analyses. These data correspond to those of previous studies in both normal and HDACi-pre-treatment culture systems of hepatogenic differentiation of MSCs [12,29]. More characteristics of mature hepatogenic differentiation of hWJ-MSCs, the mRNA expression of *G6P*, *C/EBPα*, and *CYP2B6* were different by performing significantly highest expression at the late stage of differentiation. These results were corresponding to a previous study of hMSCs differentiated into hepatocyte-like cells [11,15,34]. Therefore, these data indicate that differentiated hWJ-MSCs in the hepatic differentiation medium with NaBu pre-treatment condition demonstrated enhanced differentiation toward hepatocyte-like cells, compared to that observed in differentiated hWJ-MSCs in the hepatic differentiation medium without NaBu pre-treatment condition.

Functionally mature hepatocytes were evaluated via PAS staining and urea production assays. This study indicates typical mature hepatocyte functions, similar to those reported in previous studies, at a late stage of differentiation [11,12,13,15,16]. This study indicated that the strongest PAS-positive signals and the highest urea production were detected in the hepatic differentiation medium with NaBu pre-treatment condition of differentiated hWJ-MSCs on day 24 of the late stage of differentiation. These data indicate that differentiated hWJ-MSCs in the hepatic differentiation medium with NaBu pre-treatment condition not only express hepatic specific markers, but also the characteristic of functional hepatocytes.

Although this study provided a simple protocol and high efficiency of hepatocyte-like cells production from hWJ-MSCs accordance to previous published protocols [11,12,13,14,15,16], the molecular mechanism of NaBu on hepatogenic differentiation from stem cells with growth factors and cytokines cross talks signaling pathway in each stage of differentiation need further study. Many studies demonstrated that they used NaBu combination with growth factors and cytokines to differentiate stem cells toward successful hepatocyte-like cells, such as bFGF and BMP4 [22], activin A or Wnt3A [48] and so forth. Various concentrations of NaBu (0.5, 1, 2.5, 3, 5 and 10 mM) have different effects in many cell types of hepatogenic differentiation. Hay and colleagues (2008) exhibited that 0.5 and 1 mM NaBu supplementation with activin A enhanced hepatogenic differentiation of hESCs [21]. Varghese and colleagues (2019) also performed that at 1 mM NaBu combination with activin A and Wnt3A enhanced hepatogenic differentiation of hESCs [48]. At 2.5 and 3 mM NaBu alone or in combination with aFGF enhanced hepatogenic differentiation of mESCs [23,24]. However, supplementation of NaBu alone cannot support hepatogenic differentiation of MSC-like cells (MAPCs from mouse and human) [42], NaBu supplementation with growth factors or cytokines may be supported differentiation of MSCs toward hepatogenic differentiation. This study, for the first time, validates the effect of NaBu with and without EGF and bFGF supplementation on endodermal, hepatoblastic, and hepatic differentiation of hWJ-MSCs because hWJ-MSCs are more primitive in origin of MSCs. Herein, at one mM NaBu supplementation with EGF and bFGF can support differentiation of hWJ-MSCs into endoderm-, hepatoblast-, and hepatocyte-like cells at gene, protein, and mature function properties. Moreover, NaBu at 1, 5, and 10 mM can stimulate primary mouse hepatocytes at gene, protein, and function levels of gluconeogenesis activity [49]. Therefore, successful hepatogenic differentiation of stem cells by HDAC inhibitors is dependent on the sources of MSCs, microenvironments, and the optimizing concentration and timing (onset and duration) of exposure. Moreover, the fragile balancing of proliferation/differentiation, biological activity (pharmacokinetic)/toxicological characteristics, and apoptosis/cell survival are required. Therefore, HDAC inhibitors may be a key role of crossing lineage boundaries and promotion of trans-differentiation into a specific lineage by means of HDAC inhibition [50].

Together, the hepatocytes obtained from differentiated hWJ-MSCs in the hepatic differentiation medium with NaBu pre-treatment condition are more functional than those derived in the hepatic differentiation medium without NaBu pre-treatment condition. Thus, hWJ-MSCs have potential benefits over other adult MSCs and are an attractive cell source for future studies on drug screening and cell-based therapies for liver diseases.

## 4. Materials and Methods

### 4.1. Chemicals and Reagents

All chemicals and reagents were purchased from Sigma-Aldrich Corporation (St. Louis, MO, USA) unless otherwise indicated.

### 4.2. Human Hepatocarcinoma (hHep G2) and NIH3T3 Cell Culture

A human hepatocellular carcinoma cell line hHepG2 (ATCC HB 8065, Manassas, VA, USA) and a mouse embryonic fibroblast cell line NIH3T3 were cultured in Dulbecco’s modified Eagle’s medium (DMEM) supplemented with 10% fetal bovine serum (FBS; Life Technologies Inc. Gibco-BRL Division, Grand Island, NY, USA), 100 μg/mL streptomycin, and 100 U/mL penicillin at 37 °C in a humidified atmosphere with 5% CO_2_. They served as positive controls.

### 4.3. Isolation and Culture of hWJ-MSCs

This study was approved by the Ethics Committee for Research Involving Human Subjects, Suranaree University of Technology as a minimum, the project identification code EC-58-39 (2 September 2015). Three human umbilical cord tissues were harvested and preserved aseptically after full-term delivery at the Suranaree University of Technology Hospital, Nakhon Ratchasima, after receiving informed consent from the patient. hWJ-MSCs were isolated and cultured per the tissue explant procedure, as described previously [26].

### 4.4. Characterizations of hWJ-MSCs

Characterizations of hWJ-MSCs (*n* = 3); hWJ-MSCs #1, hWJ-MSCs #2, and hWJ-MSCs #3; were established by Embryo Technology and Stem Cell Research Center (ESRC)’s laboratory (Suranaree University of Technology) and analyzed by growth kinetics [51], immunofluorescences of the expression of MSC markers (CD73^+^, CD90^+^, CD105^+^, and CD34^−^) and multipotency were conducted using our previous protocol [26]. Moreover, hepatogenic differentiation of hWJ-MSCs from three different cords was also induced by using the previously modified standard protocol for 21 days [11]. They were characterized by immunofluorescences of the expression of hepatic markers (alpha-fetoprotein; AFP and albumin; ALB) and Periodic acid-Schiff (PAS) staining before the best one of hWJ-MSCs was chosen for further study.

### 4.5. Analysis of Cytotoxicity

One thousand hWJ-MSCs were re-plated and cultured in 96-well culture plates (SPL life sciences, Gyeonggi-do, Korea) in culture medium for 6 h to allow for attachment to the substratum. NaBu cytotoxicity was assessed through its addition to the culture medium at 0, 1, 2.5, 5, and 10 mM. All cultures were maintained at 37 °C for 72 h in a humidified atmosphere of 5% CO_2_. The effects of NaBu on cell viability were quantified through a 3-(4,5-dimethylthiazol-2-yl)-2,5-diphenyltetrazoliumbromide (MTT) assay [26].

### 4.6. Treatment with NaBu

Optimization of NaBu was assessed using real-time polymerase chain reaction (RT-PCR), immunofluorescence and Western blot analyses of endodermal lineage differentiation of hWJ-MSCs for 3 day (Figure 8A). Briefly, hWJ-MSCs were exposed 0–5 mM NaBu supplemented with and without 20 ng/mL EGF (Peprotech, Rocky Hill, NJ, USA) and 10 ng/mL bFGF (Peprotech), in Iscove’s modified Dulbecco’s medium (IMDM), 100 μg/mL streptomycin, and 100 U/mL penicillin at 37 °C in a humidified atmosphere of 5% CO_2_ in air before hepatogenic differentiation. The primer sequences used for RT-PCR analysis of mesendodermal (*GATA6*) and endodermal (*CXCR4*, *SOX17*, and *HNF3β*) gene expression, including histone deacetylase 1 (*HDAC1*), are listed in Table 1. Moreover, the epigenetic profile of differentiated hWJ-MSCs was characterized via the analysis of inhibition of HDAC1 and acetylated histone H4 (Lys5) (H4K5^Ace^) expression during endodermal differentiation. The protocols for RT-PCR, immunocytochemistry and Western blot analyses are described below.

### 4.7. Hepatogenic Differentiation

Approximately 1.5 × 10^3^ hWJ-MSCs were seeded in 4-well dishes (Nunc, Roskilde, Denmark) for immunophenotypic examination and the same number of cells were seeded in 35-mm dishes (Corning, Acton, MA, USA) and 6-well plates (SPL life sciences, Gyeonggi-do, Korea) for RT-PCR, glycogen storage, and urea production assays. The plates were coated with 0.1% gelatin, and cells were cultured in standard growth medium until approximately 80–85% confluence. Hepatogenic differentiation was performed as described previously [11] with some modifications. Briefly, cells were cultured in IMDM without serum, supplemented with 10 ng/mL bFGF, 20 ng/mL EGF, 100 μg/mL streptomycin, and 100 U/mL penicillin, combined with or without NaBu at optimized concentrations, for 3 day (in the pre-treatment step). Subsequently, cells were induced to differentiate into the hepatic lineage via a further 2-step differentiation protocol. Step 1 (the differentiation step) consisted of IMDM without serum supplemented with 10 ng/mL bFGF, 40 ng/mL hepatocyte growth factor (HGF; Peprotech), and 5 mM nicotinamide for 7 day. Step 2 (the maturation step) consisted of IMDM without serum supplemented with 10 ng/mL OSM, 1 × 10^−8^ M dexamethasone, and 1% ITS-X for 14 day (Figure 8B). In all steps, the media were changed twice weekly.

### 4.8. Gene Expression Analysis

After 3, 10, 17, and 24 day of differentiation, total RNA was isolated from the cells, using a total RNA extraction kit (RBC Real Genomics, RBC Bioscience, Taipei, Taiwan) in accordance with the manufacturer’s instructions. RNA was then reverse-transcribed using oligo-dT primers for cDNA synthesis and iScript™ Reverse Transcription Supermix for RT-PCR (BioRad, Hercules, CA, USA). The expression of liver-specific genes (Table 1) was assessed using a Light Cycler^®^ 480 (Roche Diagnostics, Basel, Switzerland) and KAPA SYBR-Green PCR Master mix (Applied Biosystems, Carlsbad, CA, USA). A melting curve analysis was also performed to determine the specificity of the primers. A target gene expression was normalized to that of the reference gene *GAPDH* and calculated as expression fold change relative to control cells. RT-PCR was performed in triplicate.

### 4.9. Immunofluorescence Staining

Characterization for the protein expressions by immunofluorescence analysis was conducted using our previous protocol [26]. Briefly, cells at day 0, 3, 10, 17, and 24 of differentiation were fixed with 4% PFA for 30 min and blocked and permeabilized for 2 h at 37 °C with 2% bovine serum albumin (BSA), 5% normal goat serum, 3 mM sodium azide, and 0.2% triton-X-100. The cells were incubated at 4 °C overnight with the following primary antibodies: Mouse anti-human α-fetoprotein (AFP) (1:100), mouse anti-human cytokeratin 18 (CK18) (1:100; Santa Cruz Biotechnology, Dallas, TX, USA), rabbit anti-human hepatocyte nuclear factor 3β (HNF3β) (1:100; Santa Cruz Biotechnology), and rabbit anti-human albumin (ALB) (1:200; Abcam, Cambridge, UK) antibodies. Subsequently, the samples were incubated for 2 h with respective secondary antibodies. The samples were stained with DAPI and observed using a fluorescence microscope (Nikon Eclipse Ti-S, Tokyo, Japan). The expression of HDAC1, SOX17, and HNF3β after 3 day was also investigated via immunofluorescence staining, using rabbit anti-human HDAC1 (1:200; Millipore), anti-human HNF3β (1:100; Santa Cruz Biotechnology), and mouse anti-human SOX17 (1:100; Abcam) primary antibodies.

### 4.10. SDS-PAGE and Western Blot Analysis

Characterization for the protein expressions by a Western blot analysis was conducted using our previous protocol [26]. Briefly, Fifteen micrograms of extracted total protein at day 0, 3, 10, 17 and 24 of differentiation were separated via SDS-PAGE (sodium dodecyl sulfate-polyacrylamide gel electrophoresis; 12% or 18% resolving gel), followed by electro-transfer to nitrocellulose membranes (BioRad). The membranes were exposed to blocking buffer (5% skim milk in PBS with 0.1% tween-20 (PBST)) and incubated with rabbit anti-human HDAC1 (dilution 1:500; Millipore, Billerica, MA, USA), anti-human HNF3β (dilution 1:500; Santa Cruz Biotechnology), anti-human acetyl-H4K5 (dilution 1:250; Millipore), and mouse anti-human SOX17 (dilution 1:500; Millipore) primary antibodies of differentiated hWJ-MSCs for 3 days of differentiation. Moreover, they incubated with rabbit anti-human HNF3β and ALB (dilution 1:500; Santa Cruz Biotechnology), anti-human AFP and CK18 (dilution 1:500; Santa Cruz Biotechnology) primary antibodies at day 10, 17 and 24 of differentiation, and mouse anti-human β-actin (dilution 1:1000; Millipore) primary antibody as an internal control. Membranes were incubated with goat anti-rabbit or goat anti-mouse secondary antibodies conjugated with alkaline phosphatase (dilution 1:20,000 and 1:10,000, respectively), and blots were then developed using 5-bromo-4-chloro-3-indolyl phosphate/nitro blue tetrazolium (SIGMA FAST™BCIP/NBT).

### 4.11. Periodic Acid-Schiff Staining

The cells were cultured until days 3, 10, 17, and 24, fixed with 4% PFA, and oxidized in 1% periodic acid in PBS for 30 min. The samples were then washed three times with deionized (DI) water and incubated in Schiff’s reagent for 15 min. Subsequently, cells washed with DI water for 5 min and observed using an inverted microscope (Nikon Eclipse Ti-S).

### 4.12. Analysis of Urea Production

The cells from each experimental group, differentiated hWJ-MSCs at days 3, 10, 17, and 24 of differentiation, were incubated with IMDM containing 5 mM NH_4_Cl for 24 h, including hHep G2 cells as a positive control. The supernatants were subjected to a colorimetric urea assay performed in accordance with the manufacturer’s instructions (Quantichrom™ Urea assay kit, Bioassay Systems, Hayward, CA, USA). A fresh culture medium supplemented with 5 mM NH_4_Cl was used as a negative control. The samples from separate cultures were analyzed in triplicate for each group.

### 4.13. Statistical Analysis

A statistical analysis was performed using SPSS version 16.0 (SPSS Inc., Chicago, IL, USA), and data were expressed as the mean ± S.D. The differences between values were determined using a one-way analysis of variance (ANOVA), followed by Turkey’s HSD post-hoc test to compare differences between two groups. The data were analyzed using GraphPad Prism 5.0 (GraphPad Software, San Diego, CA, USA). A value of *p* < 0.05 was considered significant, whereas *p* < 0.001 was considered highly significant.

## Figures and Tables

**Figure 1 ijms-20-03016-f001:**
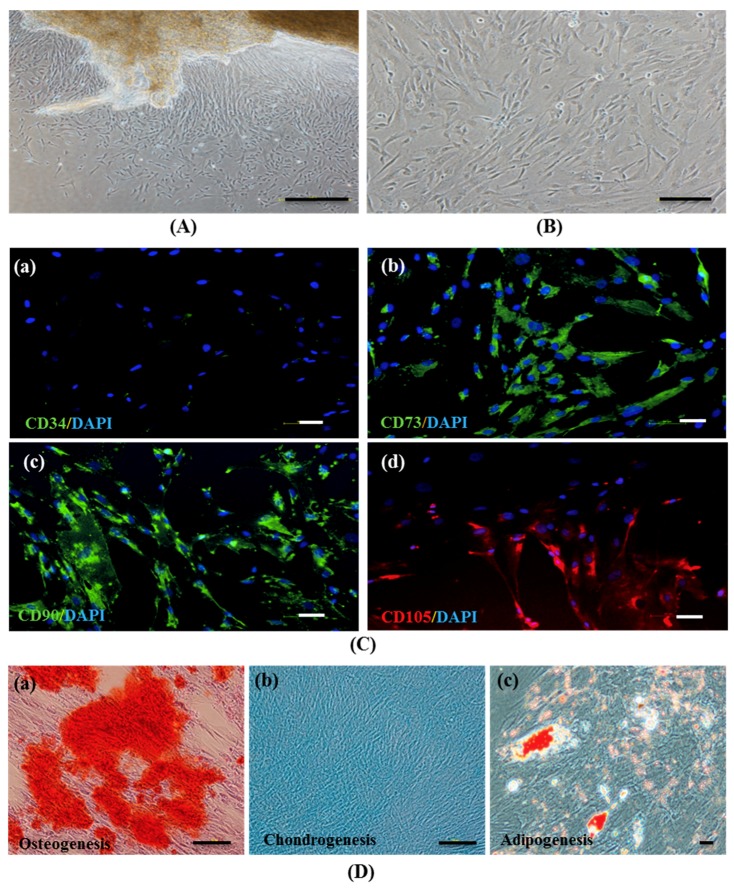
The establishment and characterization of human Wharton’s jelly–derived mesenchymal stem cells (hWJ–MSCs). (**A**) Phase–contrast microscopic images of hWJ–MSCs were expanded from Wharton’s jelly tissue and (**B**) hWJ–MSCs at 80% confluence. (**C**) Representative images of the immunophenotype of hWJ–MSCs, as assessed for CD34 (**a**), CD73 (**b**), CD90 (**c**), and CD105 (**d**) staining. (**D**) Multi–lineage differentiation potential of hWJ–MSCs after 21 days, evaluated via Alizarin Red (osteogenesis) (**a**), Alcian Blue (chondrogenesis) (**b**), and Oil Red O (adipogenesis) (**c**) staining. (Original magnifications = 40×, bar = 200 μm (**A**), 100×, bar = 100 μm (**B**,**C** (**a**–**d**), and **D** (**a**,**b**)) and 200×, bar = 50 μm (**D** (**c**)).

**Figure 2 ijms-20-03016-f002:**
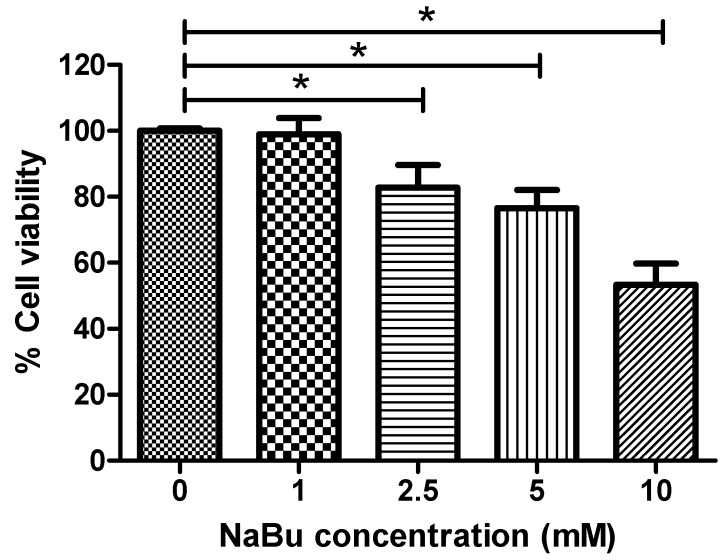
The effect of sodium butyrate (NaBu) on human Wharton’s jelly–derived mesenchymal stem cells (hWJ–MSCs) cytotoxicity. hWJ–MSCs were cultured with 0–10 mM NaBu for 3 days in 96–well plates. The cell viability was assessed via MTT assays. The data are shown as means ± SD. ** p* < 0.05.

**Figure 3 ijms-20-03016-f003:**
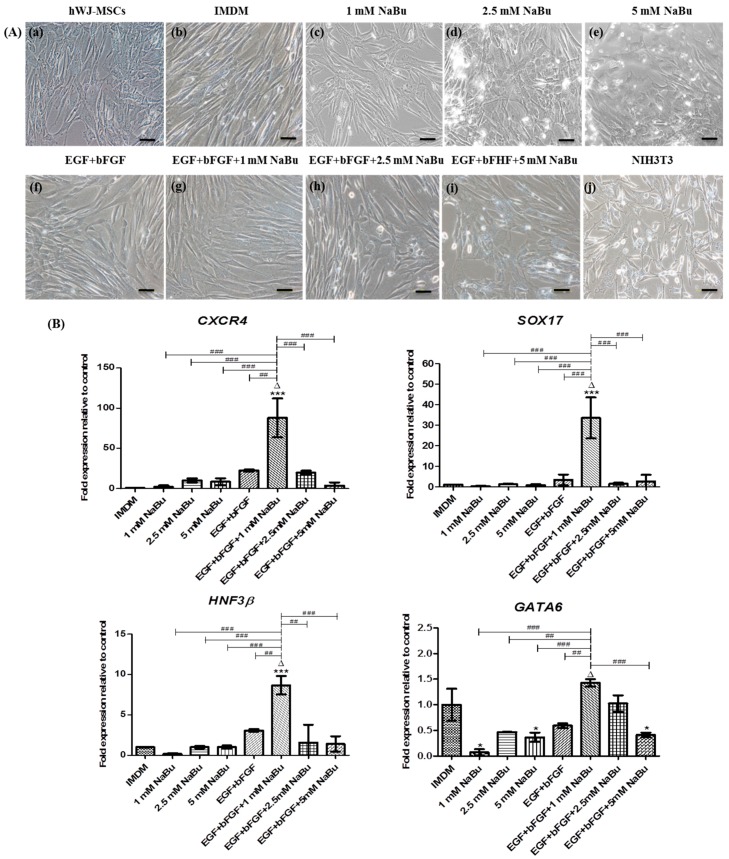
The morphological changes and real–time polymerase chain reaction (RT–PCR) analysis of mesendodermal and endodermal specific gene expressions of human Wharton’s jelly–derived mesenchymal stem cells (hWJ–MSCs) for 3 days of differentiation. (**A**) hWJ–MSCs were cultured with 0–5 mM sodium butyrate (NaBu) with and without epidermal growth factor (EGF) and basic fibroblast growth factor (bFGF) supplementation for 3 days. Phase–contrast microscopic images of hWJ–MSCs morphology changing after exposing 0–5 mM NaBu with and without EGF and bFGF supplementation for 3 days (**b**–**i**). hWJ–MSCs and NIH3T3 cells were used as negative and positive control cells (**a**,**j**). (Original magnifications 200×, bar = 50 μm). (**B**) RT–PCR analysis of definitive endoderm specific gene expression of *CXCR4*, *SOX17*, and *HNF3β* and mesendoderm specific gene expression of *GATA6* after 3 days of pre–treatment. Gene expression was normalized to *GAPDH* and was shown as expression fold–change relative to that of control cells. The experiments were performed in triplicate. The data are shown as mean ± SD, * *p* < 0.05, and *** *p* < 0.001 compared to the control group. ^∆^, represents the significantly highest mRNA expression (*p* < 0.001) compared to all other groups. ^##^
*p* < 0.01, and ^###^
*p* < 0.001 compared to the significantly highest mRNA expression group.

**Figure 4 ijms-20-03016-f004:**
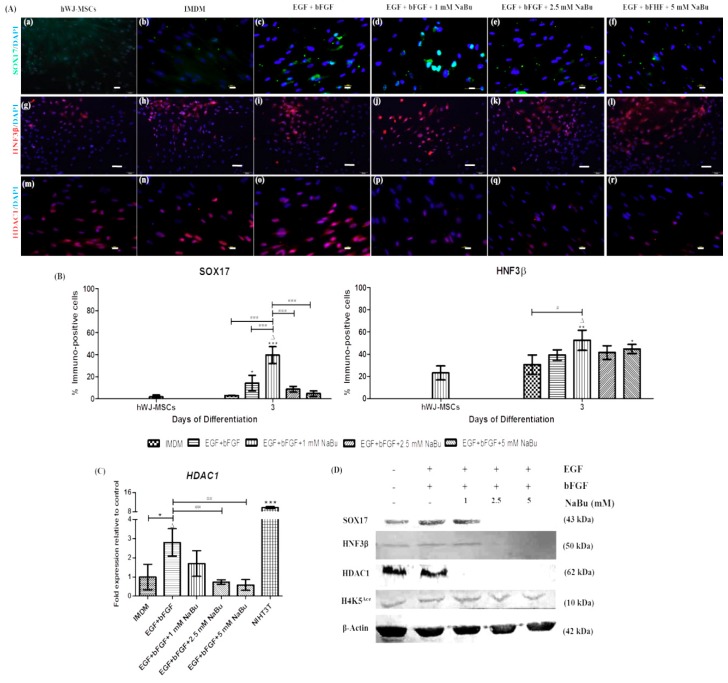
The influence of sodium butyrate (NaBu) along with the epidermal growth factor (EGF) and basic fibroblast growth factor (bFGF) on the definitive endodermal differentiation capacity of human Wharton’s jelly–derived mesenchymal stem cells (hWJ–MSCs). (**A**) Immunofluorescence staining of the definitive endoderm and histone deacetylase type 1 (HDAC1) proteins of hWJ–MSC for 3 days of differentiation. Representative indirect immunofluorescence staining for SOX17 (**a**–**f**), HNF3β (**g**–**l**), and HDAC1 (**m**–**r**) after NaBu (0, 1, 2.5, and 5 mM) pre–treatments combined with EGF and bFGF for 3 days. (Original magnifications 400×, bar = 10 μm (**a**–**f**,**m**–**r**) and 200×, 50 μm (**g**–**l**). (**B**) Immunopositive cells ratio of the definitive endoderm–like cell–derived hWJ–MSCs in various NaBu induction groups and the controls for 3 days. Overall, the 1 mM NaBu condition displayed significantly higher immunopositive cells in all definitive endodermal markers than other conditions. SOX17 and HNF3β represent definitive endodermal markers. All treatments were enumerated in five different fields (*n* = 5). The data are shown as mean ± SD, * *p* < 0.05, ** *p* < 0.01, and *** *p* < 0.001 when compared to the control group. ^∆^, represents significantly higher protein expression (*p* < 0.001) than that in all other groups. ^#^
*p* < 0.05, and ^###^
*p* < 0.001 compared to the highest protein expression group. (**C**) Real time–polymerase chain reaction (RT–PCR) analysis of *HDAC1* gene expression after 3 days of pre–treatment. The gene expression was normalized to *β–ACTIN* and was calculated as an expression fold change relative to control cells. NIH3T3 cells were used as a positive control. The experiments were performed in triplicate. The data are s hown as the mean ± SD, * *p* < 0.05 compared to the control group. ^∆^, represents significantly higher mRNA expression (*p* < 0.01) compared to all other groups. ^##^
*p* < 0.01 compared to the significantly highest mRNA expression group. (**D**) Western blot analysis of SOX17, HNF3β, HDAC1 and H4K5^Ace^ protein levels after pre-treatment for 3 days; β–actin was used as an internal control.

**Figure 5 ijms-20-03016-f005:**
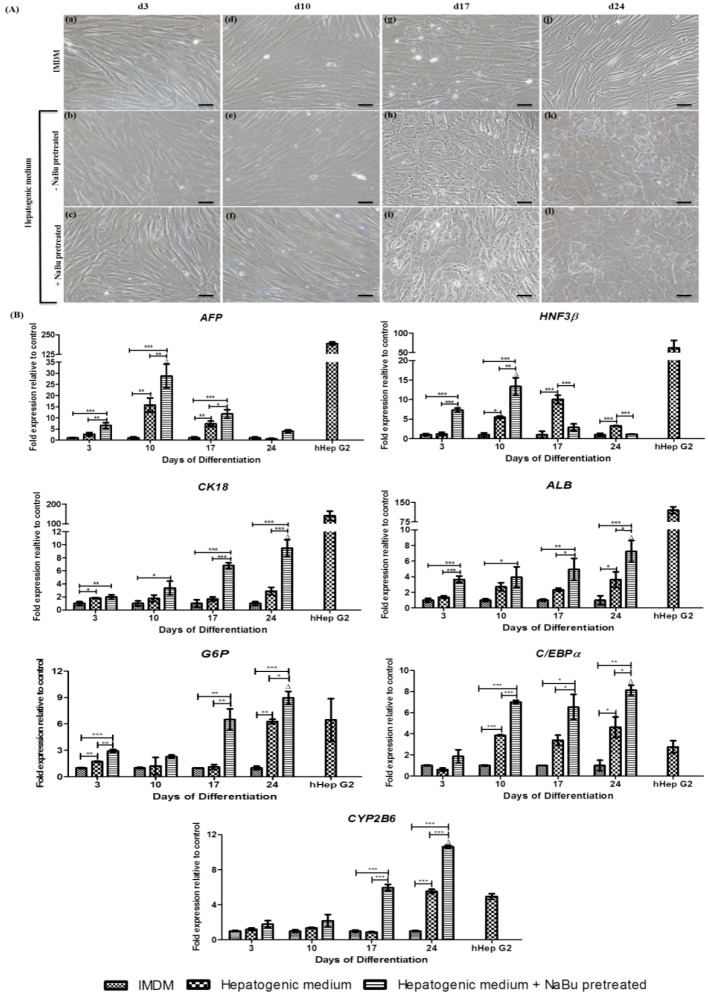
The influence of sodium butyrate (NaBu) along with epidermal growth factor (EGF) and basic fibroblast growth factor (bFGF) on the hepatogenic differentiation capacity of human Wharton’s jelly–derived mesenchymal stem cells (hWJ–MSCs). (**A**) Hepatic–like cell features of the differentiated hWJ–MSCs for 24 days of differentiation. hWJ–MSCs and differentiated hepatocyte–like cell morphology, with and without NaBu pre–treatment and EGF and bFGF supplementation at day 3 (**a**–**c**), 10 (**d**–**f**), 17 (**g**–**i**), and 24 (**j**–**l**), was observed using a phrase–contrast microscope, following the differentiation period. (Original magnifications 200×, bar = 50 μm). (**B**) A real–time polymerase chain reaction (RT–PCR) analysis of hepatic–specific gene expression in differentiated hWJ–MSCs with and without NaBu pre-treatment with EGF and bFGF supplementation for 24 days of differentiation. *AFP*, *HNF3β*, *CK18*, *ALB*, *G6P*, *C/EBPα*, and *CYP2B6* of gene expression was normalized to *GAPDH* and was calculated as an expression fold–change relative to the control cells. hHepG2 cells were used as a positive control. The experiments were performed in triplicate. The data are shown as the mean ± SD; * *p* < 0.05, ** *p* < 0.01, and *** *p* < 0.001 compared to the control and the significantly highest mRNA groups. ^∆^, represented the significantly highest mRNA expression (*p* < 0.01) compared to all other groups.

**Figure 6 ijms-20-03016-f006:**
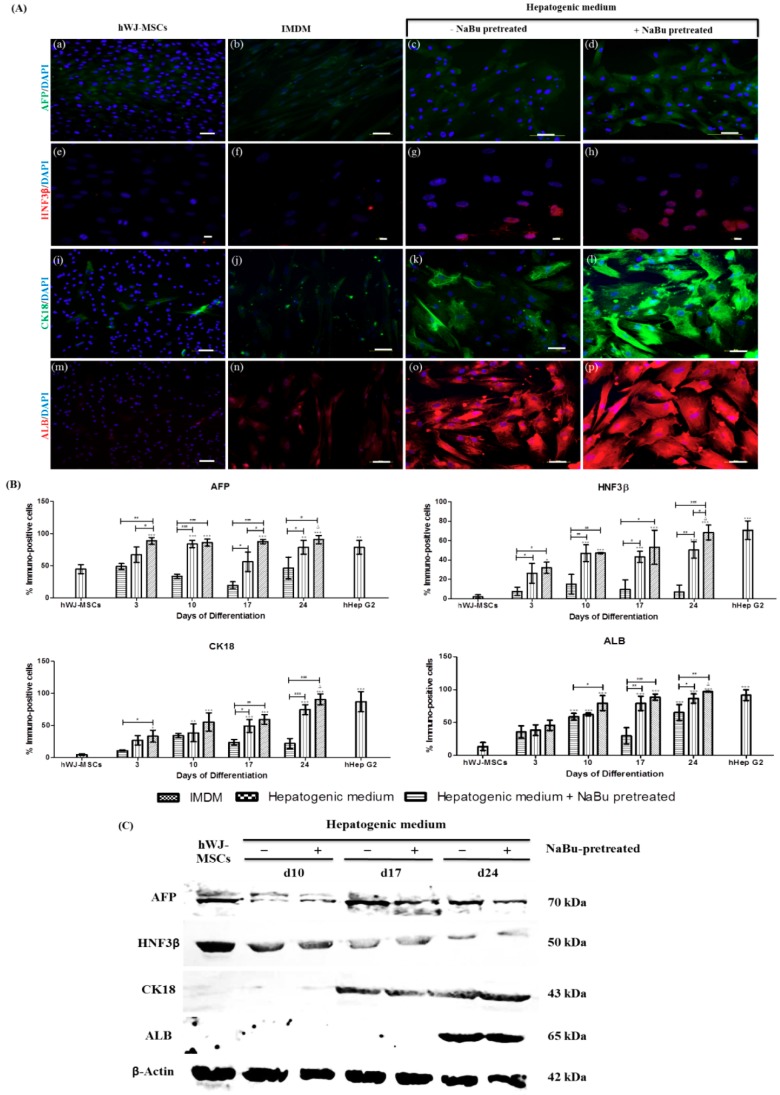
Immunofluorescence and a Western blot analysis of hepatic–specific protein expressions in differentiated human Wharton’s jelly–derived mesenchymal stem cells (hWJ–MSCs) with and without sodium butyrate (NaBu) pre–treatment, in combination with epidermal growth factor (EGF) and basic fibroblast growth factor (bFGF) supplementation on day 24. (**A**) hWJ–MSCs and differentiated hWJ–MSCs with and without NaBu pre–treatment and EGF and bFGF supplementation were stained with specific antibodies for AFP (**a**–**d**), HNF3β (**e**–**h**), CK18 (**i**–**l**), and ALB (**m**–**p**). (Original magnifications 200×, bar = 50 μm (**a**–**d**,**i**–**p**) and 400×, bar = 10 μm (**e**–**h**). (**B**) Immunopositive cells ratio of hepatocyte–like cell–derived hWJ–MSCs in NaBu induction groups and the controls for 24 days. Overall, the 1 mM NaBu condition displayed significantly higher immunopositive cells in all hepatic markers than other conditions. AFP and HNF3β represent hepatoblast–like cells or early stage of hepatic markers. CK18 and ALB represent hepatocyte–like cells or mid–late stage of hepatic markers. All treatments were counted in five different fields (*n* = 5). The data are shown as mean ± SD, * *p* < 0.05, ** *p* < 0.01, and *** *p* < 0.001 compared to the control group. ^∆^, represents significantly higher protein expression (*p* < 0.001) than that in all other groups. ^#^
*p* < 0.05, and ^###^
*p* < 0.001 compared to the significantly highest protein expression group. (**C**) A Western blot analysis of hepatic markers after 24 days of differentiation (AFP, HNF3β, CK18, and ALB) after 10, 17, and 24 days of induction were examined; β–actin was used as an internal control.

**Figure 7 ijms-20-03016-f007:**
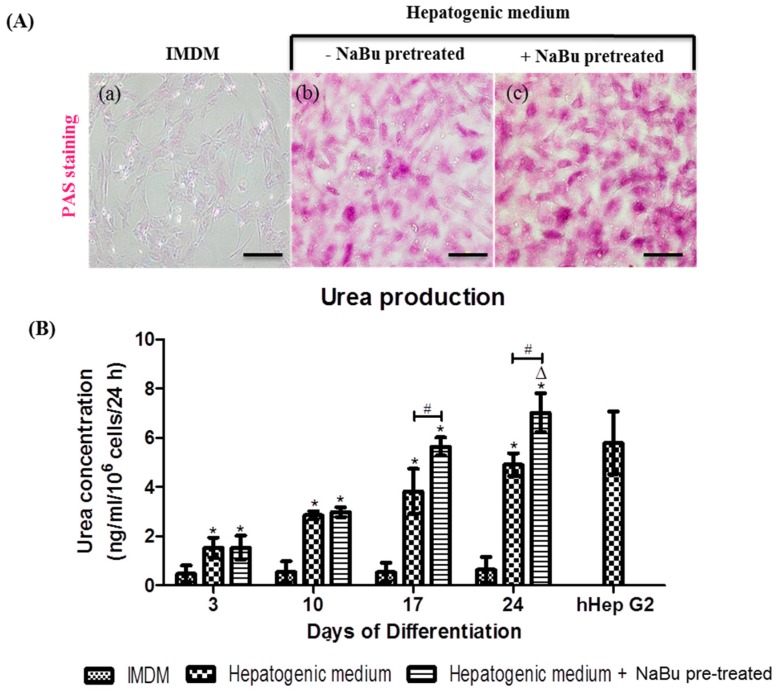
The functional evaluations of differentiated human Wharton’s jelly–derived mesenchymal stem cells (hWJ–MSCs) with and without sodium butyrate (NaBu) pre-treatment with epidermal growth factor (EGF) and basic fibroblast growth factor (bFGF) supplementation. (**A**) Glycogen storage was characterized via Periodic acid–Schiff (PAS) staining following the differentiation period in differentiated hWJ–MSCs with and without NaBu pre–treatment and EGF and bFGF supplementation on day 24 (**a**–**c**). (Original magnifications 100×, bar = 100 μm). (**B**) Urea production was assessed using a colorimetric assay in the differentiated hWJ-MSCs with and without NaBu pre–treatment and EGF and bFGF supplementation after 24 day of differentiation. hHepG2 cells were used as a positive control. The data are shown as the mean ± SD; * *p* < 0.05 compared to the control group. ^∆^, represents significantly higher urea production (*p* < 0.05) than that in other groups. ^#^
*p* < 0.05 compared to the significantly highest urea production group.

**Figure 8 ijms-20-03016-f008:**
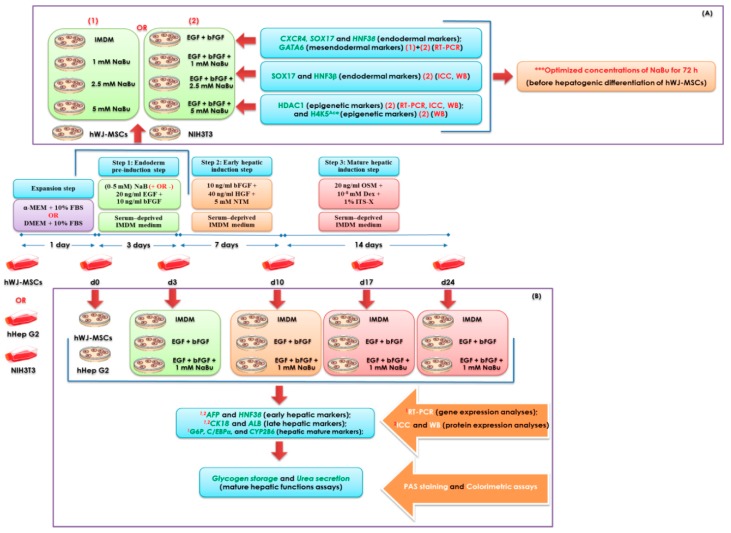
A schematic diagram of the human Wharton’s jelly–derived mesenchymal stem cells (hWJ–MSCs) hepatogenic differentiation protocol. (**A**) A schematic representation of the optimization of NaBu used for hepatogenic differentiation of hWJ–MSCs by assessing epigenetic statuses and endodermal differentiation of hWJ–MSCs for 3 days. (**B**) The representation of hWJ–MSCs differentiation toward hepatic lineage by using modified 3–step differentiation protocol and their characterizations.

**Table 1 ijms-20-03016-t001:** Primers used for RT–PCR analysis.

Genes	Primer Sequence (5′→3′)	Annealing Temperature(°C)	Product Size (bp)	References
*SOX17*	F: GGCGCAGCAGAATCCAGA	65	61	[47]
	R: CCACGACTTGCCCAGCAT			
*CXCR4*	F: ACTACACCGAGGAAATGGGCT	60	133	NM_003467.2 This study
	R: CCCACAATGCCAGTTAAGAAGA			
*GATA6*	F: CCATGACTCCAACTTCCACC	62	214	[52]
	R: ACGGAGGACGTGACTTCGGC			
*AFP*	F: CTTTGGGCTGCTCGCTATGA	60	131	NM_001134 This study
	R: GCATGTTGATTTAACAAGCTGCT			
*HNF3β*	F: CCTACTCGTACATCTCGCTCATC	65	69	[42]
	R: CGCTCAGCGTCAGCATCTT			
*CK18*	F: TCGCAAATACTGTGGACAATGC	60	171	NM_199187.1 This study
	R: GCAGTCGTGTGATATTGGTGT			
*ALB*	F: GCACAGAATCCTTGGTGAACAG	65	101	[53]
	R: ATGGAAGGTGAATGTTTCAGCA			
*G6P*	F: TCAGCTCAGGTGGTCCTCTT	62	291	[54]
	R: CCTCCTTAGGCAGCCTTCTT			
*C/EBPα*	F: ACAAGAACAGCAACGAGTACCG	65	129	[53]
	R: CATTGTCACTGGTCAGCTCCA			
*CYP2B6*	F: GTGATCTTTTGTGTCTGGTTGC	62	138	NM_000767.4 This study
	R: GATAGACGGAAGCAGTAGGAAG			
*HDAC1*	F: CCAAGTACCACAGTGATGACTACATT	65	135	NM_004964.2 This study
	R: AGAACTCAAACAGGCCATCAAA			
*GAPDH*	F: TGCACCACCAACTGCTTAGC	60	87	[53]
	R: GGCATGGACTGTGGTCATGAG			
*β-ACTIN*	F: TCACCACCACGGCCGAGCG	60	350	X00351 This study
	R: TCTCCTTCTGCATCCTGTCG			

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
