# Peer review of "Enhanced Hepatogenic Differentiation of Human Wharton’s Jelly–Derived Mesenchymal Stem Cells by Using Three-Step Protocol"

_ijms, 2019, doi:10.3390/ijms20123016_

Round 1

Reviewer 1 Report

major points:

-) in figure 4 authors displayed the expression of indicated genes at different conditions. it seems that not the all bands belong to the same loading control, if so it would be better that authors provide individual loading controls. Even in the shown case, loading control is not equally loaded, as the first band is at least 100 times more loaded in comparison to the last band, which could be reflected in above mentioned bands too.

-) in figure 6, could the author check the expression of two MSC genes such as cd90 and cd105, to be sure of the purity of the hepatic cells after being treated for about 20 days with cocktails.

-) fig 6c, the same comment as mentioned for figure 4.

-) in figure 7a, authors showed the PAS staining to confirm the differentiation completion in presence of NaBu. However, PAS is a generic staining and even the MSCs are strongly positive for it. If authors search for solid tumors with MSC origin such as Ewing sarcoma, osteosarcoma etc..., they will see that PAS counts as one of the approved stainings. Here authors need to show other confirmative staining or WBs of both MSC and hepatocyte to show the negativity for MSC and positive staining for hepatocyte.

minor points:

-) title need to be more informative and shorter

-) one repetitive word 'alternative' in introduction section (line 56).

-) in line 66 authors wrote "this approach for the differentiation of hWJ-MSCs into functional hepatocyte-like cells that express ". Differentiation is per se already happened. This process would be called Re-differentiation.

-) it has been published that NaBu has toxic impact on the cell survival, how authors would justify it? (please have a look here: J Cell Physiol. 2018 Apr;233(4):3578-3589. doi: 10.1002/jcp.26214)

-) in figure 5b the legend is written 'realtive fold expression, do the authors mean relative fold expression as the expression level has been relatived to control?

Author Response

Dear....Reviwer I

             This is the file to answer your the doubted question.

             Sorry for delayed response.

       Best regards,

    Wachira   Panta

Reviewer 2 Report

General comments. 

This is a manuscript describing that sodium butyrate enhanced hepatic differentiation of mesenchymal stem cells (MSCs) derived from Wharton's jelly. Authors established new MSC-line and analyzed the effect of sodium butyrate on the differentiation of MSCs. However, there are >30 previous studies showing that Wharton's jelly-derived MSCs differentiate into hepatocyte lineage. Several studies showed that the treatment with sodium butyrate regulates the differentiation of stem cells. The novelty of this study is slightly unclear. There are several technical problems in this study.

Specific points. 

#1. There are >30 previous studies showing that Wharton's jelly-derived MSCs differentiate into hepatocyte lineage. Several studies showed that the treatment with sodium butyrate regulates the differentiation of stem cells. In order to clarify the novelty of this study, authors should show the molecular mechanisms of the treatment of MSCs with sodium butyrate. Authors should show a comprehensive assay, such as microarray, in the differentiation of MSCs treated with sodium butyrate. 

#2. How many lines of MSCs were established from Wharton's jelly in this study?

Are the phonotypes of such cells identical? Authors should show/describe this point and comment to the reproducibility of the data of different MSC lines.

#3. Figure 2. Is there a difference of cell viability between Wharton's jelly-derived MSCs and MSCs derived from other sources?

#4. Figures 3-5. Authors should show a comprehensive assay, such as microarray, in the differentiation of MSCs treated with sodium butyrate. Authors should assess the differentiation markers of hepatocytes, cholangiocytes, and mesenchyme, which include the expression of HNF4alpha and CYP activity. 

#5. Figure 6. The quality of photographs for immunoblots (especially for albumin and beta-actin) was poor. The presentation of graphs in Fig. 6B was too complicated. Please replace and rearrange the data.

#6. The additive effects of sodium butyrate on hepatic differentiation of MSCs were unclear. ALB and urea production was not changed by the treatment with sodium butyrate in Figure 6C and 7B. How about the other functions as hepatocytes?

Author Response

Dear....Reviwer 2

        This is the file answer your doubted question.

        Sorry for delaying response.

       Best regards,

    Wachira   Panta

Round 2

Reviewer 1 Report

Title need to be corrected: by should be removed in the title and everywhere in the text authors have written it in the same context.

Reviewer 2 Report

Authors revised the points suggested by the reviewers.